# Reduction of Ultrafiltration Membrane Fouling by the Pretreatment Removal of Emerging Pollutants: A Review

**DOI:** 10.3390/membranes13010077

**Published:** 2023-01-08

**Authors:** Jianguo Zhang, Gaotian Li, Xingcheng Yuan, Panpan Li, Yongfa Yu, Weihua Yang, Shuang Zhao

**Affiliations:** 1School of Chemistry and Materials Science, Jiangsu Normal University, Xuzhou 221116, China; 2School of Mechanics and Civil Engineering, China University of Mining and Technology, Xuzhou 221116, China

**Keywords:** ultrafiltration, pretreatment, emerging pollutants, membrane fouling

## Abstract

Ultrafiltration (UF) processes exhibit high removal efficiencies for suspended solids and organic macromolecules, while UF membrane fouling is the biggest obstacle affecting the wide application of UF technology. To solve this problem, various pretreatment measures, including coagulation, adsorption, and advanced oxidation, for application prior to UF processes have been proposed and applied in actual water treatment processes. Previously, researchers mainly focused on the contribution of natural macromolecular pollutants to UF membrane fouling, while the mechanisms of the influence of emerging pollutants (EPs) in UF processes (such as antibiotics, microplastics, antibiotic resistance genes, etc.) on membrane fouling still need to be determined. This review introduces the removal efficiency and separation mechanism for EPs for pretreatments combined with UF membrane separation technology and evaluates the degree of membrane fouling based on the UF membrane’s materials/pores and the structural characteristics of the cake layer. This paper shows that the current membrane separation process should be actively developed with the aim of overcoming specific problems in order to meet the technical requirements for the efficient separation of EPs.

## 1. Introduction

In the context of the shortage of water resources worldwide, water pollution has become a serious threat to human life and health [1,2]. With the development of industrial technology, many emerging pollutants (EPs) are gradually coming to be detected in water environments, in addition to conventional pollutants. Among them, microplastics (MPs), antibiotics (antibiotic resistance genes, ARGs), and antibiotic-resistant bacteria (ARB) are regarded as three typical EPs [3]. Plastic products introduce MPs into water, soil, and air in the form of direct release or environmental degradation [4,5,6]. MPs have even been detected under extreme environmental conditions, such as those found on Mount Everest and in Antarctica [7,8,9]. A higher proportion of MPs being introduced into water systems will make it difficult for MPs to degrade and form biofilm coatings in organisms, thus affecting the biological immune system. In addition, according to incomplete statistics, by 2019, more than 4.95 million people had died from diseases caused by bacterial drug resistance [10], and ARB poses a major threat to the safety of drinking water through the spread of contaminated or improperly purified water sources [11]. Many researchers have expressed ongoing concern about the toxicological and synergistic effects of MPs and have expressed the need to explore their environmental risk.

Ultrafiltration (UF) processes represent a water purification method that is able to achieve relative energy savings, and which is widely used in biological protein separation, medical treatments, and wastewater purification [1,12,13,14]. Based on the interception effect of UF membranes on organics with different molecular weights, these processes can be adopted to separate macromolecular organics such as proteins, nucleic acids, polysaccharides, humic acids (HAs), and bacteria in wastewater purification [12,13,15,16,17,18,19]. Yasui et al. reported that adopting UF strategies to remove the median of viruses can achieve 2.9 log, much lower than the indicators of human enteroviruses [19]. UF still has incalculable potential in the separation field in the future due to its advantages, such as good retention performance, high recovery rate, fast filtration speed, wide application environment, low cost, no phase transfer, and no introduction of new impurities [13,20,21]. Adsorption, electrostatic repulsion, and size exclusion are considered to be the three main mechanisms of UF processes [22]. Macromolecules over 500 Da in size can be effectively removed (≈70%) [23], and plasmids with genetic material more than 1 kDa in size can produce a retention effect of 99% [24]. However, the retention capacity of smaller molecules (200~800 Da) such as antibiotics is very limited [25,26]. Because UF only relies on adsorption and weak retention to produce the effect of removing antibiotics, it is not considered to be a technology destructive to antibiotics [26]. Michael et al. evaluated the initial UF concentration for different antibiotics (clarithromycin, sulfamethazole, erythromycin, ampicillin, ofloxacin, trimethoprim, and tetracycline, 100 μg·L^−1^), revealing different removal efficiencies ranging from 19% (trimethoprim) to 95% (clarithromycin) [26].

The core material of ultrafiltration processes is the filter membrane. Inorganic ceramics and PVDF membranes with similar pore diameters have similar removal effects on EPs. However, the removal efficiency of ceramic materials on eARGs is better, which is due to the adsorption effect of the non-polar oxide surface and the deformability of the materials themselves [27,28,29,30]. Rigid ceramics have a stronger ability to maintain their pore size. In contrast, PVDF materials have a certain degree of toughness. In addition, the staggered state between pores will further increase the compromise of the latter on the interception effect. As one of the main consumables, the filter membrane significantly changes the membrane flux and retention capacity due to the adsorption, blockage, corrosion, and other effects of pollutants [12,18,31], resulting in an inability to achieve 100% reuse, which seriously affects the costs of UF processes when they are in actual use [32,33]. Membrane pollution can be divided into reversible membrane pollution and irreversible membrane pollution. The distinction for membrane pollution classification is generally considered to be whether or not the clogged pores can recover the original membrane flux after [21]. In addition, the membranes that trap EPs (such as MPs, etc.) also result in environment pollution, and the degradation method is also worthy of attention. It is generally believed that HAs, polysaccharides, and proteins are the main contributors to membrane pollution, and that the concentration of biopolymers can predict the behavior of sewage on membrane pollution [21,33,34,35,36,37,38].

UF membrane separation technology has shown good retention or removal capacity for various EPs in practical employ. The membrane pore size between 1~100 nm can intercept macromolecules exceeding 0.5 kDa [23,25,26]. Multi-unit filtration is considered to be one of the strategies that is able to effectively improve the membrane separation effect [39,40,41,42]. The corresponding screening effect of different size ranges on molecular weight can also be used to calculate the molecular weight distribution of organic substances [43,44]. The removal effect of ARGs mainly depends on the pore size of the membrane [45], and compared with spiral and round DNA molecules, linear molecules with similar molecular weight often have more appropriate escape ability [22]. It was found that nearly 77% of plasmids were able to escape at the aperture of 40 nm, and the removal rate of plasmids improved to 99.9% when the molecular weight of 20 kDa was intercepted [46,47]. Coelho et al. synthesized a photocatalytic ultrafiltration membrane (Ce-Y-ZrO_2_/TiO_2_) on ZrO_2_ SiC carrier by the sol–gel method, and found that the molecular retention was about 19 kDa, which is equivalent to a 6 nm aperture [48]. Riquelme Breazeal et al. also reported that when the retention molecular weight was controlled at 1 kDa and 10 kDa, the removal rates of eARGs were 4.2 log and 3.6 log, respectively [49].

Membrane pollution as a result of MPs mainly takes place in the form of surface scaling, surface wear, and pore blockage [6,50]. The pore size of the UF membrane determines that MPs below 1 nm cannot be effectively blocked. Traditional UF membrane materials usually exhibit a single electrical property, while electrostatic adsorption is not universally applicable for removing MP particles with two electrical properties at the same time [51]. In view of the state stability of MPs, the space restriction effect generated by the filter membrane avoids the deposition of fibers on most membrane channels. Large and fibrous MPs are mainly distributed in the filter cake layer after membrane interception, and MPs separated from leachate will continue to accumulate in sludge [52]. Size trapping is the key mechanism for removing MPs, while electrostatic repulsion and hydrophobic interaction mainly affect the adsorption rate of MPs on the membrane surface [53]. Blocking of pores and scaling on the surface are the main forms of membrane pollution caused by MPs. The blocking of internal pores, the formation of the filter cake layer, and the filtration of filter cake layer are the space–time processes of membrane pollution caused by MPs [53]. Mitigating membrane fouling should be considered a necessary problem to be solved in the development of ultrafiltration process. Usually, it is used in combination with coagulation, advanced oxidation processes (AOPs), such as adsorption and other pre-treatment processes, to achieve the expectation of improving the pollutant removal efficiency and of mitigating ultrafiltration membrane pollution in aquatic purification projects [13,27,38,54]. Although membrane separation processes such as UF, nanofiltration (NF), and reverse osmosis (RO) do not introduce new impurities [16,55], these technologies always exhibit the selective removal of EPs due to their charged properties [56,57]. Therefore, researchers have endowed traditional UF membrane materials with the ability to adsorb and retain EPs by modifying or using the composite [57].

At present, emerging pollutants such as MPs, ARGs, intracellular organic matter (IOM), algal organic matter (AOM), heavy metal ions (HMIs), and ARB in aquatic environment have been widely investigated and reported by researchers and environmentalists. Coagulation, adsorption, advanced oxidation, and ultrafiltration processes can all be applied for the removal of emerging pollutants. However, no single process is able to achieve sustainable and ideal results, while the use of one or two of the first three process in combination with ultrafiltration is considered to be a more promising process model. This effectively alleviates membrane pollution and strengthens the removal efficiency of emerging pollutants, while also improving the service life and industrial value of membrane materials. Many researchers have proposed many innovative solutions considering the membrane filtration mechanism, influencing factors, environmental risks and other comprehensive factors as shown in Figure 1. For example, membrane surface modification, pore size regulation, synthesis of self-cleaning materials, reduction of disinfection by-products and other forms. This review mainly, but not exclusively, limits itself to taking MPs, ARGs, extracellular polymerics, AOM, HMIs, ARBs, and other EPs as the target pollutants and summarizes the energy efficiency and action mechanism of a combined UF treatment process on EPs. Through previous work, our understanding of membrane pollution control strategies was deepened, and the design of membrane materials and the impact of mitigation pretreatment technology on membrane pollution were described based on this existing foundation. This work aims to point out directions for the efficient and personalized design of membrane materials and for the structural optimization of future engineering applications.

## 2. EPs and Pretreatment Processes

### 2.1. Environmental Risks of EPs

At present, the three representative pollution sources in EPs are MPs, ARGs, and ARB [3,45,58]. The particle size of MPs in the traditional sense is usually maintained at a size level lower than 5 mm [4,5,6]. Although MPs are generally considered to be chemically inert, their small particle size, high surface energy, wide distribution range, and ability to carry different kinds of charges on their surfaces endow them with the ability to easily adsorb and be combined with various pollutants [4,5,6,51]. Color, particle size, shape, material and surface functional groups are the main factors that affecting the adsorption capacity and binding state of MPs [7,59]. Liu et al. believed that the surface electronegativity of aged MPs was enhanced, which strengthened the electrostatic repulsion. Compared with unaged MPs, it increased the possibility of adsorbing hydrophilic substances [50]. Wang et al. reported that the efficiency of chlorination or ozone oxidation treatment for MPs removal is not ideal, and the incomplete degradation process may lead to the increase in the number and concentration of MPs [60]. In addition, MP intake makes it difficult for MPs to degrade and form biofilm coatings in organisms. At the same time, the chemical reactions that take place in the immune system may be carried out in the environment of the organism to accumulate toxins, thus affecting the biological immune system [7,8,9]. It was pointed out that in water environments with HAs, the adsorption of tetracycline and other resistance genes on MPs would be enhanced due to the complexation mechanism [61]. As widely used medical drugs, antibiotics have made important contributions to the prevention and treatment of diseases [62]. However, only a small portion of antibiotics is ingested and utilized by humans and animals, and most antibiotics are transferred to the environment in different ways [10,11,63,64,65]. Li et al. collected and analyzed the influent water of four waste water treatment plants (WWTPs) and detected 17 different types of antibiotics with high sulfonamide contents, with average concentrations of 0.13~15.33 ng·L^−1^. Using sulfamethazine (SMT, 84.6 ng·L^−1^), sulfamonomethoxine (SMM, 5.0 ng·L^−1^), sulfaquinoxaline (SQX, 105.1 ng·L^−1^), and sulfamethoxazole (SMZ, 42.5 ng·L^−1^) as examples, the detection frequency was determined to be 100% [65]. Antibiotics in biological wastewater are unable be completely removed, leading to the widespread emergence and spread of antibiotics in the environment and leading to the selective formation and retention of ARGs [3,65,66].

As an important place at which EPs converge, WWTPs have received extensive attention and reports. Shi et al. revealed that microbial colonies are the main hosts accommodating ARGs spread in WWTPs. The main biological pollutants in the effluent of fermentation processing plants are Firmicutes (Figure 2A). The abundance of ARGs in the aqueous phase is generally higher than that in the sludge phase (Figure 2B). No matter in the aquatic phase or sludge phase, the distribution abundance of environmental bacteria reflects a significant difference with the difference of pesticide equipment (Figure 2C). The proportion of transferable ARGs in the aqueous phase and sludge phase accounted for 43.6 ± 16.2% and 44.8 ± 18.0% of the total abundance of their respective phases, respectively [67]. Kang et al. sampled five WWTPs and found that the accumulation of ARGs can effectively promote the production of ARB. Liu et al. evaluated the recovery ability of different types of DNA using a continuous ultrafiltration scheme, and the results revealed that the DNA recovery efficiency of total cells and *E. coli* was 96.5 ± 18.5% and 88.0 ± 2.0%, respectively [41]. The recovery efficiencies for extracellular DNA of different lengths were also different, significantly depending on the fragment length (10.0 and 4.0 kbp (kilobase pair), 62.2~62.9%; 1.0 and 0.5 kbp, 38.8~44.5%) [41].

Algae are widely distributed in natural surface water, with many organic components and rich hydrophilic functional groups. The proteins, polysaccharides, lipids, and nucleic acids produced through the metabolic processes of algal life activities are collectively referred to as -AOM [31]. Different from traditional inorganic substances, algal cells usually attach organic layers to the surface, and improper treatment may directly lead to serious membrane pollution [31,55]. It is generally believed that membrane pollution caused by algae cells is not easy to remove by physical cleaning, and irreversible pollution is mainly caused by the adsorption of extracellular organics [68,69]. Moreover, the common HMIs, (e.g., Cu^2+^, Fe^3+^, Zn^2+^, etc.) found in aquatic environments have attracted the attention of an increasing number of researchers because of their serious biological toxicity and immune function damage [56,57], all of which are discussed in this review.

### 2.2. Removal Mechanism of EPs by Pretreatment Processes

Compared with using a single water purification technology, using a combined UF pretreatment strategy is considered to be a more efficient water treatment scheme [70]. Although part of the membrane flux can be recovered after physical or chemical cleaning, the increase in transmembrane pressure (TMP), permanent deactivation of some channels, and degradation of the material’s own performance will directly affect the membrane life [54,71,72]. TMP can generate differential driving force, and the high-voltage driving force can shield the electrostatic repulsion between the membrane and pollutants to some extent [16]. In addition, when the porosity is low due to membrane fouling, the driving force generated by a TMP that is too high may directly lead to membrane material damage [73,74]. However, after pretreatment process, HAs and sodium alginate (SA) with macromolecular structure can bridge organic matter, or have adsorption or net capture effect with suspended particles, and then further develop into large sized flocs or micelles [38]. Macromolecules carrying pollutants can be more smoothly blocked at one side of the UF membrane, reducing the retention ratio of pollutants in the channel, and concentrating pollutants in the filter cake layer [37,74]. Therefore, the structure can be effectively separated from the membrane surface after physical flushing with pure water [74].

Coagulation, as an efficient and low-cost process, is considered to be one of the most effective pre-treatment schemes for mitigating membrane pollution [38,70,75]. Coagulation preferentially combines negatively charged particles and organic macromolecules in the water through the electric neutralization mechanism and then continuously captures pollutants in the form of adsorption bridging and net sweeping, with the growth of the floc particle size accelerating the phase separation process of pollutants [21]. For the removal of ARGs, HAs, and other complex water source pollutants, the advantages of the coagulant itself can be considered to a greater extent, and the organic matter and turbidity of the UF influent can be reduced effectively [42,58,76]. Floc properties such as size, structure, resilience, and shear resistance are the main reasons for the reduction in irreversible membrane pollution [38,58,77], and sedimentation time is one of the key indexes affecting floc characteristics [13,78]. Compared with direct UF (Figure 3, left), PACl hydrolysates can effectively capture suspended particles and organic macromolecules, which gather in the filter cake layer on the UF membrane surface (Figure 3, middle). Inorganic organic coupling coagulation process can further improve the density of the filter cake layer (Figure 3, right), and effectively inhibit the exposure of membrane pores to pollution sources.

As a pretreatment process, adsorption has the outstanding advantages of low cost, high efficiency, and no driving force, but also has the disadvantages of high regeneration costs and easy secondary release [79]. Through the combination of adsorption and ultrafiltration (A-UF), the process originally belonging to two units can be integrated into one process, reducing the footprint of the water purification unit [18,20,25,74]. Carbon-based materials have the characteristics of high-specific surface area and many adsorption sites. They can adsorb various pollutants through hydrophobic interaction, hydrogen bonding, and π–π stacking [20]. Due to the characteristics of the large specific surface area and the low particle size of pretreatment materials, improper use will lead to the deposition of adsorption materials on the surfaces of UF membranes, which will further aggravate membrane pollution [18,25,74]; therefore, A-UF has high requirements regarding the combination of materials and the actual operation process.

AOPs are a technology involved in the oxidative degradation and mineralization of macromolecular unsaturated organic compounds through the generation of hydroxyl radicals, sulfate radicals, superoxide radicals, and other intermediates with photoelectric catalysis or strong oxidants [27,79]. AOPs can degrade organics in water sources efficiently, especially in small molecule organics that cannot be removed by coagulation, which are characterized by high universality, fast response, a high degree of mineralization, and low product toxicity (or non-toxic products). However, compared with adsorption and coagulation/flocculation, the demand for energy is higher [27,79,80]. In addition, residual oxidants will accelerate membrane aging and cause irreversible material consumption [35,75]. Electrochemical catalysis or strong oxidant catalysis will inevitably bring about problems such as incomplete degradation and pollutant residues [15,39,81]. The transformation products show high toxicity potential and low biodegradability in their chemical structures and may be carcinogenic or mutagenic under specific circumstances [81].

AOPs attack the active sites in the long chain of polymers by producing strong oxidative free radicals (e.g., ·OH, SO_4_^−^, S_2_O_8_^−^ and O_2_^−^, etc.), so that MPs gradually degraded into smaller molecules of organic matter [82,83]. In the degradation process of color MPs, organic dyes released along with the decomposition of polymers. Therefore, the ability to catalyze the degradation of dyes also needs to be included in the evaluation system. It is considered relatively safe when the release rate of dyes does not exceed the degradation rate [83]. In addition, the catalytic system and reaction conditions also affect the degradation products. Using appropriate catalysts to attack specific target sites to obtain decomposition products with higher expected values has become a promising direction [84]. Lu et al. adopted the electro catalytic advanced oxidation strategy and selected sodium dodecyl sulfate (SDS) as the surfactant to enhance the adsorption of strong oxidants on the MPs surface. As a result, the catalytic degradation rate was about 1.35~2.29 times than that of the boron doped catalyst alone [85]. Among a series of degradation products produced in the 72 h process, alkane cracking and oxidation products were detected, which proved that AOPs could effectively attack and realize the further oxidation process of saturated alkanes. MPs trapped on the membrane surface are transferred to the sludge phase through backwashing (reversible) or chemical cleaning (irreversible), and then decomposed in the form of AOPs or thermal cracking [86,87,88,89]. AOPs can not only be used in the pretreatment process of UF, but also can be used as an effective way to degrade MPs deposited in sludge.

## 3. Contribution of Pretreatment Scheme for EP Removal

As shown in Table 1, different pretreatment processes may have different removal effects on various EPs. Coagulation or adsorption can enrich EPs and remove them from wastewater in the form of phase separation [11,80], which is the most popular membrane strategy for pollution mitigation in the current scheme, but the enriched pollutants still need to be further treated [75]. AOPs can directly degrade natural organic matter (NOM) to form low-toxicity or non-toxic small molecule products, and it is universal for low-concentration EPs [75,80]. However, the small-molecule products generated from the degradation of different EPs may be diversified, so AOPs may lead to the biological risk of effluent by-products. The removal efficiency of the pretreatment scheme is not only related to the ability to fix/decompose EPs, but also related to the concentration of EPs in the influent.

### 3.1. Coagulation/Flocculation

Although common Al-, Fe-, and Ti-based coagulants perform the adsorption, capture and fixation processes of pollutants with similar mechanisms, they can have different coagulation effects [31]. By forming large-sized flocs, coagulation distributes pollutants more widely in the filter cake layer on the membrane surface, preventing small-sized flocs and free-macromolecular organic substances from entering the inner part of the channel to cause irreversible membrane pollution [100]. Coagulation processes are unable to effectively adsorb organic matters with a molecular weight less than 1 kDa [35,127]. However, Al/Fe deposited on a UF membrane surface can continuously and effectively adsorb eARGs from wastewater, and its retention effect can be improved to 7-log. Xu et al. pointed out that, under different scenarios, the promotion of coagulants on ultrafiltration efficiency is obviously different. For example, under alkaline or neutral conditions, poly aluminum calcium (PACl) shows the best coagulation efficiency and membrane pollution mitigation capability. Under acidic conditions, metal hydrolysis is incomplete, making it easy for small flocs to form, which is conducive to improving membrane filtration flux. Compared to Fe and Al, Ti salt can generate a stronger positive adsorption field due to higher charge density and promote the aggregation of algae and AOM, forming larger flocs. In addition, because a large number of metal hydrolysates encapsulate algae, avoiding excessive contact between the organic layer and the membrane surface, it has the best performance in mitigating membrane pollution [31].

Based on the traditional coagulation method, the additional application of polysaccharide coagulant aids can significantly reduce the demand for coagulants and can, meanwhile, enhance the floc performance (size and crushing strength) and water purification efficiency of coagulation [38,79,128]. Inorganic–organic compound strategies can further reduce the dependence of the subsequent separation process on the UF membrane, thus prolonging the service life of the UF membrane [13,79,129]. Cui et al. adopted potassium aluminum sulfate (PAS) and chitosan (CTS) as inorganic coagulant and organic coagulant aids, respectively, and combined them with corn straw as auxiliary biomass in a coagulation-ultrafiltration (C-UF) process. They found that the interception and adsorption generated by the system can effectively inactivate or remove HMIs and antibiotics [130]. Compared with single coagulation or biomass adsorption, the membrane flux can be doubled, and the formation of membrane pollution can be significantly inhibited, avoiding additional flushing caused by the recovery of membrane flux [130]. However, the removal effect of EPs by an inorganic–organic compound coagulant will be significantly affected by the changes in pH, ionic strength, humic acid concentration, and other factors [131]. The coagulation mechanism under the coexistence of EPs and other pollutants needs further research and verification [17,132,133]. It has been reported that a quaternary amine-group-grafted chitosan flocculant can effectively exert the “breaking-cell-wall” effect, enabling the removal efficiency of *E. coli* to reach 99% [134]. However, the release of IOM and the production of nitrogen-containing disinfection by-products (N-DBP) mean that quaternary ammonium groups are considered unsafe. Compared with the quaternary ammonium group, when starch is used as the precursor surface to graft quaternary phosphine salt, it can achieve a higher bactericidal effect (99.4%), without the risk of N-DBP, and can prevent the diffusion of aromatic proteins and other IOMs after the death of bacteria [135,136,137,138]; in addition, compared with alum, it significantly improves membrane pollution mitigation [138,139].

Figure 4a,b show the scaling phenomenon on the surface of polystyrene (PS) particles intercepted directly by the UF process. The percolation effect of a large number of PS particles deposited on the membrane surface barrier pores is also the main cause of TMP increase. Figure 4c,d show the filter cake layer structure produced by removing PS using C-UF coupling process. PS is mainly distributed inside the filter cake layer, and the loose structure is conducive to the long-term retention of water flux. The TMP produced by removing MPs with UF is positively related to the influent concentration. The flocs produced by aluminum-based coagulant (26 mM) form a loose filter cake layer on the membrane surface, effectively alleviating the direct contact between MPs and UF membrane materials [140]. Compared with the single UF process, the C-UF process inhibits the increase in TMP by 85%. Through the rotation of membrane filtration unit components, the water inlet side transits from laminar flow to turbulent flow. The formation of stronger shear force is an important driving factor that causes the filter cake layer to fall off and the continuous dynamic accumulation of MPs, which alleviates the deterioration of membrane fouling caused by the continuous accumulation of filter cake layer [140]. In addition, as high molecular organic flocculants, polyacrylamide (PAM) and SA can produce gel-like flocs, and UF continuous filtration significantly increases the specific filtration resistance (SFR) and TMP. The formation of gel layer is different from membrane pore plugging and traditional hydrophobic filter cake layer. The introduction of low dose PAM (20 mg/L) can effectively reduce SFR by reducing negative Zeta potential and system uniformity, while high dose PAM (50 mg/L) enhanced the negative Zeta potential and homogeneity of the system, which promoted the SFR to be raised again. With the increase in PAM dose, SFR showed a “V” change trend [141]. The contribution of filter cake layer to the mitigation of membrane pollution is not always positive. The structure of filter cake layer produced by NOM is directly related to the components. The membrane pollution caused by protein can be reduced by HAs or polysaccharide. The coexistence of HAs and polysaccharide will form a typical sandwich structure, leading to increased membrane resistance and reduced flux. The presence of HAs or polysaccharide can cause more uniform distribution of pollutants in the filter cake layer and lower membrane fouling [142].

### 3.2. Adsorption

Adsorption as a pretreatment combined with UF is diverse. Zhang et al. established a support layer by depositing activated carbon fiber (ACF) on the surface of a polyethersulfone (PES) membrane. ACF combined with UF processes showed an effective removal effect (76%) in the range of 10^1^~10^6^ ng/L for the steroid hormone concentration [20], making up for the defects of traditional UF processes in the selective permeation of small molecule organics. Cao et al. reported that by concentrating extracellular polymer on the filter cake layer on the UF side, HMIs (Pb^2+^, 94.8%; Cu^2+^, 88.9%; Cd^2+^, 89.2%) can be effectively adsorbed, and extracellular polymer achieved a recovery rate of 85.5%, indicating that it is an effective strategy to alleviate membrane pollution [16]. Extracellular polymer can form a stable filter cake layer (11.6 µm) on the UF membrane surface. Pb adsorption or deposition can significantly reduce the thickness of the filter cake layer (9.2 µm). Figure 5 clearly reflects the rearrangement and contraction of filter cake structure caused by HMIs. Michael et al. used granular activated carbon (GAC) to make contact with antibiotics for 90 min before combining it with the UF process to achieve almost complete removal [26]. However, macrolide antibiotics show higher membrane repulsion due to their high hydrophobicity, and ofloxacin shows anisotropic repulsion in different pH scenarios, which may be caused by the transformation of material charge and electrostatic interaction with the environmental pH value [26].

The effect of adsorption on different kinds of EPs is obviously discrepancy. Li et al. evaluated the treatment efficiency of five kinds of antibiotics in actual sewage by investigating WWTPs using ozone/biological activated carbon (O_3_/BAC) for oxidation/adsorption in a combined UF strategy [65]. The final removal efficiency was in the following order: macrolides (96.8%) > chloramphenicols (96.3%) > sulfonamides (95.5%) > quinolones (84.1%) > antifungal pharmaceuticals (66.7%). The removal efficiency of antibiotics using the conventional UF process was lower than 40%, but the ozone/powdered activated carbon (O_3_/PAC) pretreatment produced a removal efficiency of 80.1%, and the total removal efficiencies achieved were 92.3% (sulfonamides) and 94.8% (macrolides). The results reveal that oxidation and adsorption can effectively remove most sulfonamides, and oxidation can produce high removal rates of macrolide antibiotics [65]. UF processes do not contribute much to the process of removing antibiotics, and their main role is to filter out large activated carbon particles after they adhere to pollutants [65]. From another perspective, oxidation/adsorption and other pretreatment schemes are feasible and effective for inhibiting UF membrane pollution. Zambianchi et al. designed a polysulfone graphite oxide hollow fiber membrane (PSU-GO-HFs) that has both adsorption and ultrafiltration capabilities. The adsorption effect of membrane materials was evaluated with ciprofloxacin (CIPRO), HMIs (Pb (II), Cu (II) and Cr (III), and perfluoroalkyl substances (PFASs, C4-C13) as the target pollutants [143]. The results revealed that the removal effect of PSU-GO HFs on CIPRO, HMIs, and PFASs was better than that of commercial GAC, and there was almost no secondary release of EPs in the high-speed flow process [143]. Khaliha et al. found that the adsorption effect of rich defective GO was nearly six times that of commercial GAC by comparing the adsorption effect of rich defective GO (d) and low-defective GO (b) on antibiotics and methylene blue (MB) [144]. After adsorption, the combined use of UF processes can eliminate the environmental risk of secondary pollution [144].

### 3.3. AOPs

The mode of electric-, optical-, metal-, and strong oxidant activation-driven persulfate PS has been applied by researchers for the removal or degradation of various EPs [28,133]. Photoelectrochemical catalytic oxidation is an AOPs technology that can effectively degrade low-concentration antibiotics. When combined with UF processes, it can effectively limit the spread of drug resistance and achieve the interception and inactivation of ARGs and ARBs [3,133]. Deniere et al. adopted the O_3_/PMS activation system to catalyze the degradation of trace organic pollutants (TrOCs). The amount of hydroxyl radicals generated by the ozone/peroxymonosulfate (O_3_/PMS) system is three times that of O_3_ oxidation [145]. When 12.3 mg/L O_3_ is used, the degradation effect of O_3_/PMS is 24% higher than that of O_3_ oxidation [145]. Mousel et al. adopted an Ultrafiltration membrane bioreactor (UF-MBR) as the separation process to evaluate the removal effect of adsorption, oxidation, and AOPs (UV/H_2_O_2_) systems. Using 17 typical target drugs (1000~30,000 ng·L^−1^) as examples, the three pre-treatment methods achieved a removal efficiency of more than 80%, and the residual concentration of most pollutants was less than 10 ng·L^−1^ [80]. Li et al. adopted a combined coagulation flocculation persulfate (CF-PS) strategy to effectively remove organics with a molecular weight more than 50 kDa, and the removal rate of the dissolved organic carbon (DOC) was 41.3%. When PS is used alone, the removal rates of 3–50 kDa and DOC lower than 3 kDa are 62.7% and 40.3%, respectively. CF pretreatment enhances the removal of medium- and high-molecular-weight lysine and tryptophan, and PS can remove humic organics and extracellular polymers more thoroughly [43]. Zhou et al. reported a new strategy of using Ag/TiO_2_/GO as AOP catalysts and PVDF for synergistic removal, efficiently inactivating low-concentration ARGs in target wastewater through interfacial adsorption and oxidation degradation [133]. The study also pointed out that the main reason for the deactivation of ARGs is the mineralization of bases and the phosphodiester. The GO structure contains a large number of sp^2^ carbon skeletons and oxygen-containing functional groups. Through π-π interaction and the hydrogen bond adsorption of ARGs, the bases in the molecules are completely inactivated, and the results after mineralization are not reversed [133].

The beginning of oxidative degradation is usually activated by energy drive, and then different free radicals are generated according to the specificity of the oxidation system. Lumbaque et al. proposed a design idea using an inner-tube photoreactor to improve the removal efficiency of peroxydisulfate (PDS) through photolysis and photocatalysis. Photolysis and photocatalysis pathways include the following: (1) PDS dissociates S_2_O_8_^2−^ for direct oxidation; (2) PDS driven by UV to break the homologous O-O bond and generate SO_4_^•−^ after activation; (3) UV-excited H_2_O/S_2_O_8_^2−^ system to generate ^•^OH/S_2_O_8_^•−^; (4) UV light-driven activated O_2_/S_2_O_8_^2−^ system reduced to O_2_^•−^/SO_4_^•−^ [81]. The tubular ceramic UF membrane can be used as the carrier of the contact surface between the catalyst and the oxidant, effectively promoting PDS’ participation in the transport process of the catalytic surface and water inlet [81]. Sulfate free radicals usually undergo the electron transfer reaction, while ·OH can react through H extraction and electron transfer processes [81]. Song et al. adopted the electrolytic oxidation ceramic ultrafiltration (EO-CM) process to conduct a pseudo first-order kinetic degradation of SMZ [146]. In the oxidation time (0~60 min) and applied current (5~30 Am/cm^2^) ranges, the degradation efficiency was positively related to the two factors. In the process of electrolytic oxidation degradation ClO^−^ and HClO or Cl_2_, H_2_O_2_, and ·OH are the intermediates involved in the oxidation process. SMZ cannot be effectively removed by ultrafiltration processes, but it can be efficiently degraded during pretreatment. EO can maintain high throughput in CM processes, significantly controlling membrane pollution [146]. Krzeminski et al. adopted radiation levels of 100 and 300 mJ/cm^2^ UV (265 nm) and combined them with UF for 601 and 267 bp genetic fragments, respectively, to make the DNA content in the target effluent lower than the minimum detection limit, and this is considered to be one of the feasible technologies expected to achieve the goal of completely eliminating ARGs [24].

### 3.4. Environmental Risk Associated with Pretreatment Processes

Physical cleaning and chemical cleaning are common membrane cleaning methods, and each has its own advantages. Compared with physical cleaning, chemical cleaning is considered to be an effective way to clean irreversible pollution [15,99], and chemically enhanced backwashing (CEB) is a way to maintain membrane flux by combining physical and chemical cleaning [115]. The by-products of the CEB process mainly consist of volatile halogenated hydrocarbons (VHOC) and halogenated acetic acids (HAAs). The halogenated by-products significantly affect the quality and safety of membrane effluent [15,54]. Chemical cleaning with hypochlorite enriches proteins, polysaccharides, and amino acids increases the risk of halogenated by-product generation. The concentration of by-products is positively related to the concentration and temperature of hypochlorite. Cleaning can cause membrane flux recovery, increased pore size, decreased hydrophilicity, and increased HA-exclusion ability [15]. Using the ozone ultrafiltration (O_3_–UF) strategy generates a significant risk of carcinogenic by-products of HAAs in the CEB process [15,54]. In contrast, when C-UF uses Al as the coagulant, it will not cause carcinogenic risks to the overall water purification process, and its biological safety is higher [54]. Alpatova et al. proposed a scheme using saturated CO_2_ solution as a cleaning agent to effectively reduce TMP and to achieve the efficient removal of bovine serum albumin (BSA) [115]. In the depressurized membrane channel environment, the saturated CO_2_ solution will rapidly resolve and nucleate depending on the pore network structure to promote the removal of pollutants at the inlet side by virtue of the expansion effect generated by the bubble growth process. Membrane flux can be completely recovered using sodium alginate and SiO_2_ nanometer particulates (NPs) in the feed water [115].

Adding strong oxidants into chemical cleaning agents can further improve the membrane cleaning effect. Li et al. adopted the combined pre-oxidation UF scheme. Via the pre-oxidation of Fe (II) and Mn (II) to FeO*_x_* and MnO*_x_*, based on in situ generated FeO*_x_* + MnO*_x_* + H_2_O_2_ as the cleaning agent, a recovery effect (0.5 wt% H_2_O_2_) of the membrane flux higher than 95% can be achieved in 5 min. The use of a coagulant (PACl) promotes the membrane cleaning effect to a certain extent. MnO*_x_* can enhance the coagulation effect of PACl. Due to the different crystallinity levels of FeO*_x_* and MnO*_x_*, FeO*_x_* does not show an obvious cleaning effect [99]. Song et al. reported that using NaOH + EDTA as a cleaning agent can achieve the effective regeneration of membrane flux, and using NaClO + HCl can further improve the recovery rate of membrane flux to 99.2% [147]. Zhou et al. believed that the development of self-cleaning ability in electrochemical membranes was mainly caused by the partial mineralization or oxidation of pollutants. Based on the hydroxyl radical and superoxide radical intermediates produced in the electrochemical oxidation process, it has been proposed that there might be three ways to degrade tetracycline, as shown in Figure 6 [148]. Extracellular polymers can reduce membrane flux through enriching the membrane surface, improve water flow resistance and the interception effect, and enhance complexation adsorption. The size-interception capacity is the main mechanism through which extracellular polymers enhances the membrane filtration capacity [149]. Extracellular polymers inhibit the degree of irreversible pollution and the risk of ARG transmission risk outside of the cell through ion bridging, hydrophobic effects, and molecular chain extension.

The cost of membrane cleaning is significantly related to the cleaning agent, cleaning method and membrane pollution degree. The thickness of the filter cake layer shows a “V” trend with increasing amount of flocculating agent. Warm water is usually used to clean the UF membrane to enhance the release of pollutants in the filter cake layer, and the additional heat supply will increase the cost by at least 0.126 RMB/ton [34,150,151,152]. The cost of water purification with inorganic coagulants is almost the same as that of AOPs, and nearly half of the total cost is saved compared with PAM [153]. PAM is an organic flocculant commonly used in the field of industrial water purification. However, the produced carcinogenic monomer acrylamide (AM) during hydrolysis makes it no longer acceptable [154,155]. Organic compounds containing amino (-NH_2_) or quaternary ammonium salt will form N-DBP during oxidation [134,156,157]. In comparison, inorganic metal-based coagulants are less toxic. The most frequently reported biological toxicity is Alzheimer’s disease induced by aluminum-based coagulant residues. Therefore, the residual concentration of metal ions, e.g., Al^3+^, 0.2 mg/L (China) [158], is usually added to the water purification safety standards. Polymeric organics containing only C, H, O elements (without phenyl-) are often considered as relatively safe agents, which is also an important reason for the popularity of polysaccharide-based flocculants (e.g., SA).

The cost distribution of carbon-based materials is diversified, since the prices of GO, carbon nanotubes, graphene and biological carbon span several orders of magnitude [159,160]. However, the adsorption capacity of carbon-based materials is usually affected by temperature, which leads to dynamic changes in desorption balance and makes low reagent adsorption efficiency [161,162,163]. In addition, the surface of carbon materials combined with heavy metal ions may lose a large number of active adsorption sites, which causes the deactivation of adsorbent and is not easy to be restored again. Therefore, many carbon-based adsorbents are always used as disposable products [164]. As a common pretreatment process of UF, AOPs are regarded as an effective way to degrade EPs deposited in sludge. Photocatalysis, electrocatalysis and chemical drive are usually used to couple with strong oxidants to further enhance the offensive capability of the catalytic system [83,84,85]. The strong oxidizing free radicals make HMIs release into the water environment again by attacking the HMIs complex in the bound state on the adsorbent surface. However, AOPs, as pretreatment methods, cannot effectively immobilize HMIs [165].

## 4. Contribution of the Pretreatment Process to the Mitigation of Membrane Pollution

### 4.1. Coagulation/Flocculation

The pretreatment of pollutants with coagulants can not only effectively prevent macromolecular NOM from coming into contact with UF membranes, it can also enhance the hydrophobic components and aggregation degree of pollutants [35]. There are three main states of Al-based coagulants in hydrolysis processes: monomer aluminum (Al*_a_*), intermediate aluminum (Al*_b_*), and high-polymer aluminum (Al*_c_*) [166]. Yuan et al. believed that combining Al*_a_* and NOM to form complex Al*_a_*-NOM was the main contributor to membrane pollution, and that the content of Al*_a_* was significantly related to the degree of membrane pollution, in the range of tens to hundreds (μg/L). Compared with aluminum chloride, the use of PACl, which accounts for a relatively low proportion of Al*_a_*, can significantly alleviate the membrane pollution caused by Al*_a_*-NOM [166]. Chen et al. proved through density functional theory (DFT) calculations that the coordination cross-linking of Fe (III) with three SA terminal carboxyl groups promoted the extension of the polymer chain. SA is prone to curl and aggregate to form alginate mixtures, which can reduce the SFR of UF membranes [33]. The difference in the iron concentration will lead to changes in the surface charge, pH, and microstructure of the mixture. The increase in the iron concentration from 0 to 2.5 mM leads to the mixture transforming from a gel to granular flocs. Due to the effect of nonterminal coordination and a compressed electric double layer, SFR first increases and then decreases. The highest value appears at 0.1 mM, and SFR is about 1.65 × 10^14^ m·kg^−1^ [33].

Organic flocculant and inorganic–organic composite coagulation systems effectively overcome the problem of small floc particle size. The introduction of organic coagulant aids has greatly enhanced the mechanisms of adsorption bridging and net trapping and sweeping, while traditional coagulants play the role of electric neutralization and the adsorption and complexation of amorphous hydrolysates. In addition, a large number of negatively charged groups (hydroxyl, carboxyl, sulfonic acid, amino, etc.) in the structure of organic polymer chain substances can be used as active adsorption sites to directionally capture metal ions during adsorption [13,79,128]. Through mixing with an inorganic coagulant, a small amount of coagulant aid is introduced to significantly reduce the dosage of coagulant and to effectively improve the water purification efficiency of ultrafiltration pretreatment processes [13,76]. In the actual application of UF membrane, the change in TMP has become one of the most important parameters for evaluating the degree of membrane pollution. TMP is increased with the introduction of biopolymer with large molecular weight (MW), thus aggravating the phenomenon of membrane fouling [115,167,168]. When the water samples are treated using different coagulants in the pretreatment process, flocs with various properties can be obtained. Generally, flocs with larger sizes and higher density exhibit stronger shear resistance, which results in a thicker cake layer on the UF membrane surface during the ultrafiltration process. In this condition, the pore blocking degree is relatively smaller, and the porosity can be recovered more effectively by backwashing [169,170]. Adding an appropriate amount of non-toxic and hydrophobic natural polymer in the coagulation process can increase the hydrophilicity of the UF membrane, thus effectively alleviating the formation of membrane pollution [170]. The hydrophobicity of the modified UF membrane itself has been proved to effectively alleviate membrane pollution. When the PVDF surface is modified with an ultra-low concentration (0.01 mg/L) of moderately hydrophobic chitosan (MHC), the structure, density, thickness and other parameters of the floc layer can be significantly improved; in addition, UF membrane showed a higher void recovery after backwashing treatment [100]. Chemical cleaning is regarded as the main way to recover irreversible membrane pollution. However, strong alkali (NaOH), strong oxidant (NaClO) and strong acid (citric acid) have irreversible aging effects on membrane materials (e.g., PVDF) [171]. In comparison, alkaline cleaning agents cause the least damage to membrane materials. After 2~10 days of cleaning, the aged membrane materials will fall off to varying degrees, and the size of fragments is negatively related to the cleaning time. Fragmented PVDF directly reduces the retention performance of the membrane materials [171]. After NaClO aging for 20 days, the recovery rate of membrane flux is significantly reduced by about 60%. After aging, the hydrophilicity and permeability of membrane materials are enhanced, which makes it easier to adsorb pollutants to form an irreversible pollution layer [172]. From the point of view of intermolecular interaction, the surface free energy of the aged membrane material is enhanced, and the thermodynamic stability is higher after the pollutants are attached [172]. The irreversible process can be effectively alleviated by reducing the amount of strong oxidants (e.g., NaClO).

### 4.2. Adsorption

The scheme for establishing a pre-adsorption layer as the “protective layer” of membrane materials is to first adsorb pollutants and then block the direct contact between main pollutants and membrane materials, and this is considered to be an effective way to alleviate membrane pollution [20,25,39]. Yu et al. adopted a coupling scheme for vulcanized activated carbon (GAC), PAC, and UF. PAC was used to ensure the quality of the UF effluent. GAC alleviates UF membrane pollution and PAC deposition (<45.8%) through particle momentum and induced liquid turbulence [25,26]. After being scoured by GAC with different particle sizes, no obvious damage and morphological differences were found on the membrane surface, indicating that GAC, as a pretreated adsorption material, does not exert negative effects on the UF process due to wear, indicating that the GAC-PAC-UF coupling process has the potential to be a feasible solution for long-term industrial application (Figure 7). The GAC-PAC-UF process can reduce energy consumption costs by an order of magnitude compared with traditional UF schemes [25]. Ali et al. adopted zeolite as the adsorption material and polyethylene glycol (polyvinyl alcohol (PVA)) and cellulose acetate (CA) as the pore modifiers to increase the hydrophilicity of the material. Compared with the electrospinning scheme, the preparation of reverse membrane materials demonstrating high adsorption performance is cheaper and easier to handle. PVA can produce higher solute rejection (93%) and improve biocompatibility [12].

The hydrophobicity of adsorbent materials is also regarded as an important contributor to the fixation of organics. Liu et al. adopted a combination scheme using a porous biochar aerogel (PBA) and UF process, and the test showed excellent performance that was beyond that of most carbon-based adsorption materials (DOC, 85.9%; divalent metal ions, 70.6%) [39]. PBA has properties such as an ultra-high specific surface area, a porous structure, and a large pore volume, which is able to be specifically combined with the hydrophobic organic part, with a DOC removal rate of more than 50.2%. After adsorption pretreatment, it can effectively alleviate the direct hydrophobic interaction between the UF membrane and pollutants [39]. Shi et al. adopted nails as the adsorption material before membrane filtration, and they were able to effectively adsorb HMIs such as (V), Cd (II), and Pb (II) in water during the oxidation process of the nails, and the rust generated was also able to adsorb organic substances. When the rust falls off, a new iron surface will be exposed, and the adsorption process will continue. The fallen rust will be completely rejected by the UF film [18]. The results revealed that the thickness of the filter cake layer increased by 143%, the content of adenosine triphosphate decreased by 75%, the removal rate of HMIs exceeded 90% (iron adsorption > 80%), and the rust was able to prevent bacteria growth [18]. During 105 days of continuous operation, the membrane flux of 3.5 L·m^−2^·h^−1^ was able to be maintained, indicating this to be a low-cost and low-maintenance solution for mitigating membrane pollution [18]. Ma et al. adopted polysulfone (PSF) andPES to prepare a block copolymer composite membrane PSF-b-PES with amphoteric affinity. The PES enriched in the membrane pores can provide effective hydrophilicity, and the exclusion efficiency of BSA is greatly improved (45% → 71%) [102].

### 4.3. AOPs

All kinds of driving strategies show good degradation ability for organic matter, and are considered effective for mitigating membrane pollution. Wang et al. adopted a synergistic strategy using photocatalysis and coagulation as the pretreatment process. By degrading high-molecular-weight organics into small molecules and capturing the hydrophilic organics, proteins, and HAs in the flocs, the total UF membrane pollution was reduced by 88% [35]. Cheng et al. adopted metal activation to generate SO_4_^·-^ and ·OH in an Fe (II)/PMS system, showing atrazine to possess good degradation ability. Low-dose Fe (II)/PMS shows an aggravating membrane pollution phenomenon. Only at high doses can Fe (II)/PMS play a mitigating role [27]. Guo et al. adopted a Xe lamp to simulate sunlight irradiation and PDS to synergistically catalyze the degradation of atrazine, revealing the difference in the membrane pollution-inhibition effect at different temperature ranges (30~70 °C). Higher temperatures can effectively promote the mitigation of membrane pollution. At 70 °C, TMP can be effectively reduced by about 70% [32].

The synthesis or modification of membrane materials can also be used as an effective measure to improve membrane performance. Xie et al. synthesized Co/N carbon hollow-fiber UF membranes derived from ZIF-67. Through special catalytic sites, the application of electric fields, and size interception, the efficiency and electrostatic repulsion of the AOPs were effectively improved, and membrane fouling caused by organic pollutant deposition was effectively inhibited [173]. The strategy of applying an electric field causes AOPs to have special activity and stability, which proves that the electro-assisted membrane-based AOP process is a sustainable and feasible strategy [173]. Zhang et al. adopted a heterogeneous Fenton system to implant iron oxychloride into ceramic UF membrane pores, effectively solving the problem of AOPs. ·OH is easily quenched by NOM and other substances due to mass transfer restrictions or water components [29]. The results revealed that the pH of the system can be as high as 6.2. In the channel of UF membranes (20 nm), the effective action time of AOPs exceeds 24 h. The retention time of ·OH can be extended to 10 s, and the membrane flux can be maintained at 100 L·m^−2^·h^−1^. By exposing small molecular pollutants to ·OH, which restricts the reaction to the membrane pore channel, the rapid quenching of free radicals can be avoided effectively, improving the kinetic catalytic efficiency of AOPs. The size-trapping effect produced by UF membranes can form effective steric hindrance for macromolecules over 300 kDa. The combination strategy of FeOCl UF eliminates the formation of irreversible membrane pollution in the way outlined in [29].

The degradation ability of different oxidation systems is still a topic of concern for researchers. Chang et al. evaluated the ability of AOPs to pretreat NOM and inhibit membrane pollution using different strategies, including UV and metal-activated PS strategies, and UV/Fe (II)/PS showed the best degradation performance [79]. Additionally, using 100 μM Fe (II) and 400 μM dosages of MPs, the NOM degradable by UV irradiation for 1 h exceeded 93%, and sulfate free radicals dominated the degradation test for carbamazepine. The thinner filter cake layer and looser structure led to lower sludge resistance, an increase in specific flow rate (J/J_0_) by 139.13%, a decrease in irreversible pollution resistance by 69.94%, a reduction in total interface energy of the pollutants interacting with the UF membrane by 84.42%, and shorting of the separation distance to nearly 2 nm [79]. The oxidation of Fe (IV) can effectively remove ARGs, especially eARGs. Compared to Fe (III) and PACl, the removal effect of Fe (IV) on iARGs is relatively low [58], which may be caused by the fact that the flocs that are produced are negatively charged particles and do not settle easily [174]. On the other hand, Fe (IV) can effectively remove eARGs (>2.15 log) that are not easily trapped by UF membranes. Therefore, Fe (IV)-UF can exert a better removal effect than traditional C-UF (3.26~5.01 log). Fluorescent humus can easily cause membrane pollution and can be effectively removed, reducing the biological toxicity, while the enhanced hydrophilicity of the floc surfaces further inhibits the formation of irreversible membrane pollution [58]. Zhou et al. designed a new type of anode film material by doping a defective Zr-based metal organic framework (D-UiO-66) and conductive graphite particles into a PVDF substrate. The system was able to effectively remove a series of antibiotics (>96.6%) and had a high bactericidal rate (≈100%) under intermittent and continuous electric field supply [148]. In addition, taking BSA as the target EP, membrane flux can achieve nearly 100% recovery and can be considered to have good stability, membrane separation, and self-cleaning capacity, and has the potential to purify urban secondary sewage for a long time.

## 5. Conclusions and Expectations

As a widely used industrial water purification technology, ultrafiltration has shown excellent and stable separation effects for traditional pollutant removal. However, there are many bottlenecks in the UF process for the removal of EPs, especially due to increased membrane pollution problems. This paper systematically summarized common pretreatment processes, including coagulation, adsorption, and AOPs, that are commonly applied to reduce membrane fouling and pointed out the defects that need to be overcome in combination pretreatment–UF processes.

Coagulation processes have the characteristic of highly efficient negative electrical pollutant removal, ensuring that negative electrical pollutants are effectively fixed in the pretreatment stage and making it easier for the UF membrane to cope with two complex scenarios with different electrical properties at the same time. C-UF can capture and concentrate pollutants first, so that pollutants can be more easily trapped. A-UF establishes a protective layer along the upstream area of the UF membrane to reduce the residual concentration of pollutants at the water inlet end of the membrane material. In other words, the two processes enable the UF membrane to deal directly with water with lower pollution levels, thereby prolonging the service life of the UF membrane and reducing the frequency of membrane backwash, thus reducing the economic costs of replacing membrane materials. AOP-UF is more universal, degrading macromolecular pollutants into smaller molecules with lower toxicity. However, a more comprehensive safety assessment is required for by-products after the degradation of different target pollutants. In addition, EPs that are resistant to oxidation do not seem to be suitable for the application of AOP pre-treatment strategies. The interaction between excessive by-products and residual reactants may also bring more uncertainty to the biological safety of effluents. Many researchers have turned their attention towards loading catalytic/oxidation media in the pores of UF membranes to form a synergistic heterogeneous catalytic system with PMS (PS) to achieve efficient catalytic degradation and to eliminate irreversible pollution in the pores from the formation path. The comprehensive evaluation of the stability and service life of the membrane system used for this strategy will become a hot research topic in future.

For combined pretreatment–UF technology, the membrane material should strengthen the “self-cleaning” ability on the basis of improving material rigidity, reducing pore size, enhancing surface hydrophobicity, and strengthening corrosion resistance and electrostatic repulsion. An appropriate compromise can be achieved regarding membrane adsorption performance in order to reduce the adsorption energy of pollutants and membrane materials. The main reasons are as follows: (I) major pollutants are removed in the pretreatment process, which is equivalent to the “adsorption” link in the membrane separation process, and direct contact between this part of pollutants and the membrane surface and pores can be avoided; (II) the enhancement of hydrophobicity and the weakening of surface adsorption can significantly improve the pore recovery rate in the membrane backwashing process. “Less contact” and a “long cycle” are positive ideal solutions for alleviating UF membrane pollution.

For complex water systems with different pollution sources, a single coupling mode of pretreatment and UF is not enough to remove all pollutants. Coagulation and AOPs cannot effectively immobilize HMIs, while the efficiency of the adsorbent is usually higher. Coagulation and adsorption are not ideal methods for MPs degradation; in comparison, AOPs show satisfactory decomposition ability. Large catalytic effects greatly reduce the conversion time and operation cost. However, avoiding incomplete degradation and small molecule toxic products are always the problems need to be solved. Adsorbents cannot fix EPs efficiently and continuously, while coagulation technology, especially low-residue coagulation systems, is more trustworthy. In view of the homology between MPs and UF membrane materials, coagulation and adsorption are always used as pre-treatment methods to fix MPs in wastewater, which are milder than AOPs.

Just like “waste classification”, various EPs need to be classified according to the dominant pollutant composition to select more appropriate pretreatment technologies. This will not only help to alleviate UF membrane pollution, but also significantly enhance the removal efficiency of pollution sources. In addition, from the perspective of cycle sustainability, it seems to be a more promising development direction to degrade different kinds of pollutants into “new energy materials”, so that they can re-enter the ecological cycle, promoting social development.

## Figures and Tables

**Figure 1 membranes-13-00077-f001:**
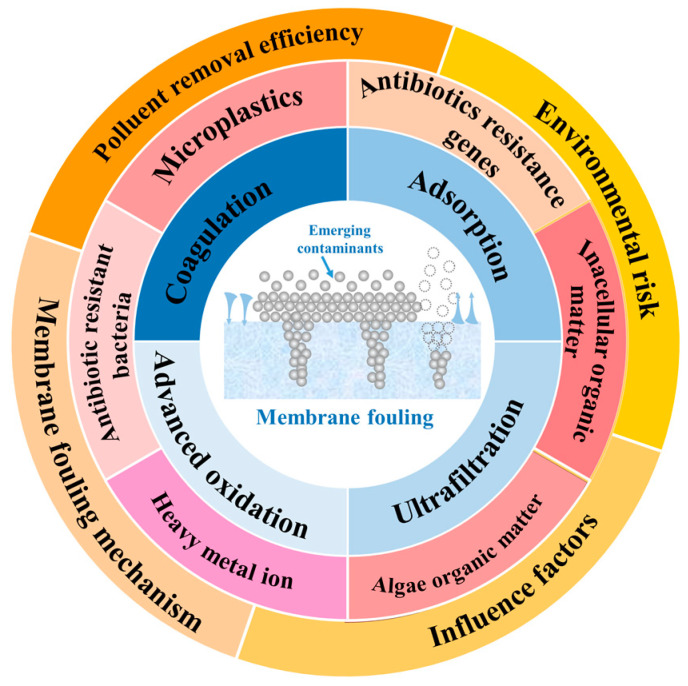
Schematic of the uses of pretreatment processes to reduce membrane fouling.

**Figure 2 membranes-13-00077-f002:**
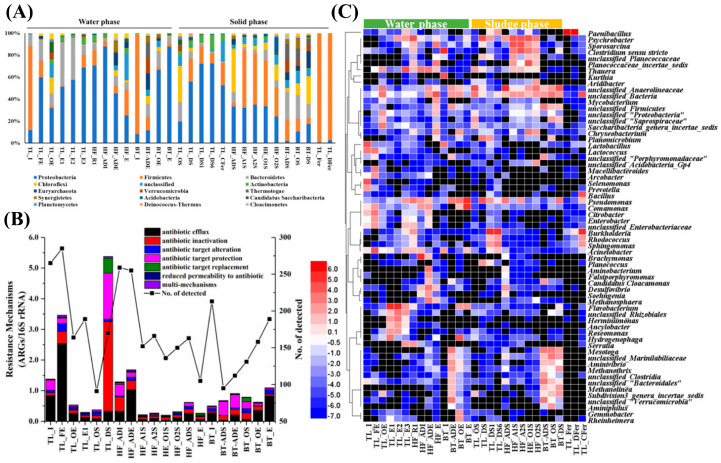
Microbial community composition in the water phase and sludge phase at the phylum level (**A**); Changes in the relative abundance of antibiotic resistance mechanisms and the number of ARGs detected at the subtype level (**B**); Heatmap of the top 10 genus in each sample (log 2 transformed) (**C**) [67].

**Figure 3 membranes-13-00077-f003:**
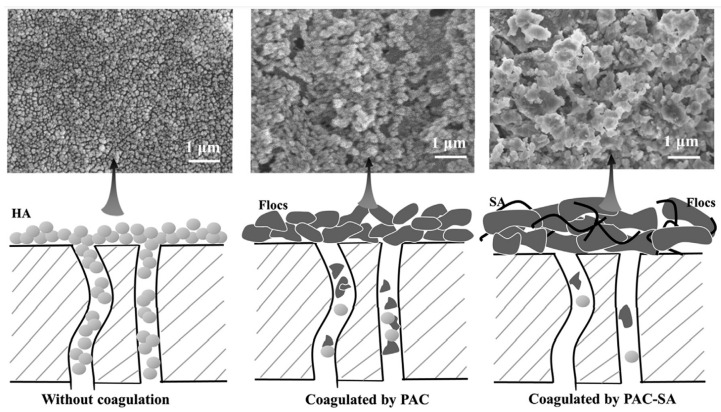
Cake layer morphology and fouling mechanism of UF membrane under different pretreatment conditions [13].

**Figure 4 membranes-13-00077-f004:**
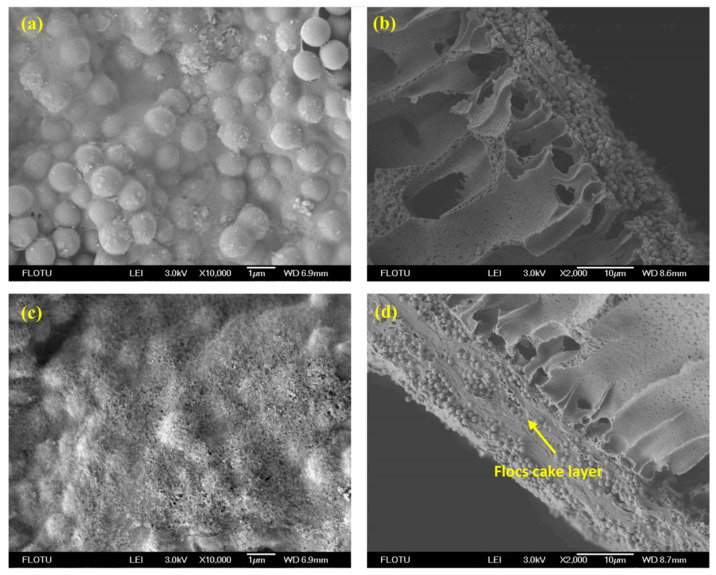
Morphology of the membrane on the 10th day when operated in the presence of 1 µm PS of 1 mg/L: (**a**) without flocs, surface, (**b**) without flocs, cross-section, (**c**) with 13 mM flocs, surface, and (**d**) with 13 mM flocs, cross-section [140].

**Figure 5 membranes-13-00077-f005:**
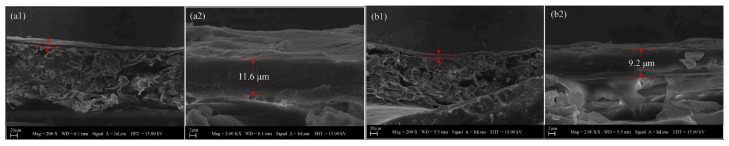
Low- (**left**) and high-magnification (**right**) scanning electron microscopy images of (**a1**,**a2**) extracellular polymer-cake and (**b1**,**b2**) extracellular polymer-cake-Pb cross-sections [16].

**Figure 6 membranes-13-00077-f006:**
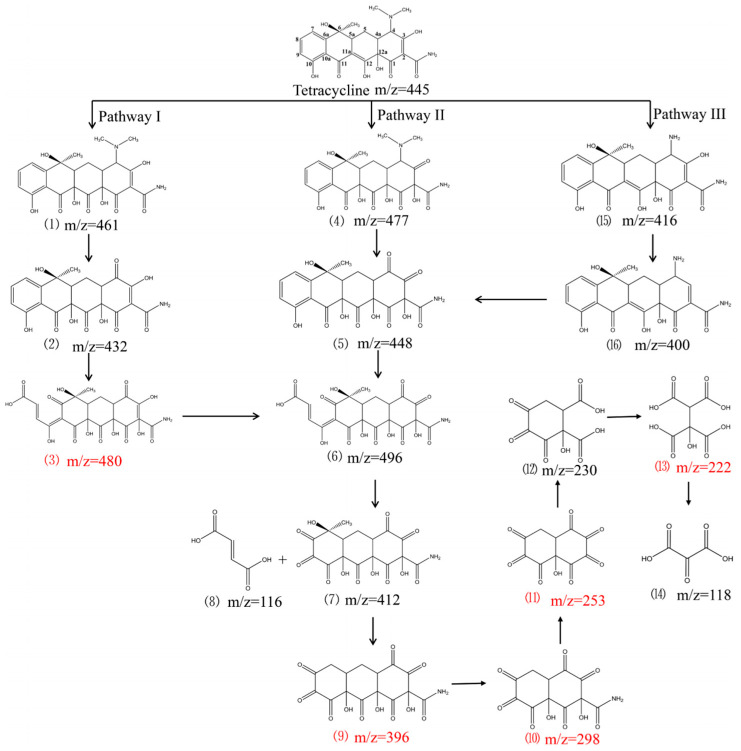
Proposed degradation pathway of tetracycline after electrocatalytic oxidation by D-UiO-66/Graphite /PVDF anodic membrane. The speculated intermediate is presented in red font [148].

**Figure 7 membranes-13-00077-f007:**
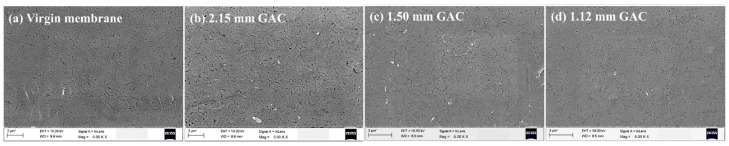
SEM images of (**a**) a virgin membrane and cleaned membranes from GAC-PAC-UF system with GAC particle diameters of (**b**) 2.15 mm, (**c**) 1.50 mm, and (**d**) 1.12 mm [25].

**Table 1 membranes-13-00077-t001:** Treatment efficiency of pretreatment combined with UF process.

Membrane Material	Pollutant Type (Concentration)	Pretreatment Method	Interception Capability	Separation Mechanism	TMP (MPa)	Membrane Flux (L/m^2^·h)	Removal Efficiency	Reference
PTFE	MPs (0.96 ± 0.46 item/L)	Coagulation/Adsorption	1 μm	Adsorption/Size retention/Electrostatic repulsion	-	-	80%	[90]
PES	MPs	Adsorption/Coagulation	0.74 μm (100 kDa)	Adsorption/Size retention/Electrostatic repulsion	0.07	-	90%	[91]
PSF	MPs	Adsorption	30 kDa	Adsorption/Electrostatic repulsion	0.128–0.32	-	70.7%	[53]
PSF	MPs (10 mg/L)	Adsorption	30 kDa	Adsorption	0.2	-	75%	[92]
PSF	MPs (10 mg/L)	Adsorption	30 kDa	Electrostatic repulsion	0.01	-	-	[93]
PES/PVP	MPs (77 ± 7.21 item/L)	AOPs	0.1 µm	Adsorption/Size retention/Electrostatic repulsion	-	-	96.97%	[94]
PVDF	MPs (1 mg/L)	Adsorption	15~25 μm (100 kDa)	Adsorption	0.0002	10	-	[95]
hollow fiber	organic	Adsorption/Coagulation/AOPs	50 kDa	Adsorption/Size retention	0.05	400	-	[96]
MBR	OMP (<50 ng/L)	AOPs	0.04 μm	Size retention	-	5	80%	[80]
Ce-Y-ZrO_2_/TiO_2_	Animal protein/HA/Phenol	AOPs/Adsorption	6 nm (19 kDa)	Adsorption /Size retention	0.1	160	-	[48]
PVDF/Co@N-C	TC (20 mg·L^−1^)	AOPs	2~80 nm	Adsorption/Size retention/Electrostatic repulsion	0.1	636.0	99.3%	[97]
Ceramic	Organic phosphorus (248 mg/L)	AOPs	-	Adsorption/Size retention	-	-	83%	[28]
Ceramic	NOM (5 mg/L)	AOPs	300 kDa	Size retention	-	100	>80%	[29]
PAA/PAH	HMIs	Adsorption	1 kDa	Adsorption	0.40–1		>85%	[98]
PES	Fe^2+^ (1.0 mg/ L)/Mn^2+^ (6.1 mg/ L)	AOPs	30 kDa	Electrostatic repulsion	1	-	>95%	[99]
HF	Norfloxacin (0.1 μg/L)/Tylosin (0.1μg/L)	Coagulation	0.03 μm (100 kDa)	Adsorption/Electrostatic repulsion	0.002	20	80~90%	[100]
(C/PVDF)	ARB/ARGs	AOPs	30–80 nm	Size retention	0.1	125	81.5%	[3]
Ceramic	HMIs/Antibiotic	AOPs/Coagulation	50 kDa	Size retention	0.04	-	-	[30]
PPSU	ARB	Adsorption	67 kDa	Electrostatic repulsion	0.276	10–150	89%	[101]
EPS	HMIs (0.02–0.16 mg/L)	Adsorption	10 kDa	Adsorption	0.2	-	94.8%	[16]
PES	HMIs (20 mg/L)	Adsorption	150 kDa	Adsorption	0.004	3.5	>90%	[18]
PSF-b-PEG	BSA	Adsorption	66 kDa	Electrostatic repulsion	0.15	59	71%	[102]
PVDF	ARGs	Adsorption	100 kDa	Electrostatic repulsion	-	-	99%	[26]
ECM	ARGs	Adsorption	-	Electrostatic repulsion	-	-	94.8%	[103]
PES	NOM	Coagulation/Adsorption	100 kDa	Adsorption/Size retention	0.06	-	-	[104]
PES	NOM (5–50 mg/L)	Coagulation/Adsorption	100 kDa	Adsorption/Size retention	0.08	-	-	[105]
ZrO_2_ mono-tubular ceramic	BSA (10.0 g/L)	Adsorption	50 nm	Adsorption/Size retention/ Electrostatic repulsion	0.15	-	86.75 %	[106]
tubular ceramic	organic compounds	Adsorption	8 kDa	Size retention	0.28–0.40	123	80%	[107]
PVDF	Casein (1 g/L)	AOPs	30 kDa	Size retention	0.10	-	-	[108]
PES	Mn (II)	Adsorption	30 kDa	Adsorption /Size retention	0.05	100	95%	[109]
PVDF	NOM	AOPs	20 nm	Size retention	0.30	60	-	[110]
PVDF	organic pollutants	AOPs	150 kDa	Size retention	0.1	-	94.9%	[111]
PVDF	NOM (20 mg/L)	Adsorption	100 kDa	Adsorption/Size retention	0.1	-	83%	[112]
RCA	Proteins (0.9 ± 0.1 mg/mL)	Adsorption	100 kDa	Adsorption /Size retention	0.1	75–132	97%	[113]
hydrophilized polyethersulphone	Organics/ protein-like substances	Adsorption	100 kDa	Adsorption/Size retention	0.06	-	79.4/84.8%	[114]
AOPs	NOM	AOPs	150 kDa	Size retention	0.0728	237	81.64%	[32]
PES	EOM (7.08 μg/mg)	AOPs	100 kDa	Size retention	0.05	-	-	[15]
PVDF	BSA	Adsorption	0.03 μm	Adsorption /Size retention/ Electrostatic repulsion	0.2	103.8	-	[115]
PES	NOM	Adsorption	100 kDa	Adsorption /Size retention	0.08	-	-	[116]
PES	EOM	Adsorption/ AOPs	100 kDa	Adsorption/ Size retention	0.10	-	-	[117]
PVDF	organic	Coagulation /Adsorption	100 kDa	Adsorption/ Size retention	0.10	320	90.06%	[118]
PVDF	BSA	AOPs	67 kDa	Adsorption/Size retention/Electrostatic repulsion	0.10	230–270	93%	[119]
FeOCl-CM	BSA	AOPs	300 kDa	Adsorption	0.10	-	95%	[120]
PVDF	NOM (5.7 mg/L)	Coagulation	100 kDa	Adsorption	0.10	-	94%	[121]
CuFeCM	NOM (20 mg/L)	AOPs	300 kDa	Adsorption	0.10	-	-	[122]
PVDF	MPs (1 mg/L)	Adsorption	100 kDa	Adsorption/Size retention/Electrostatic repulsion	0.20	10	-	[98]
PES	HMIs	Adsorption	50 kDa	Electrostatic repulsion	-	-	94.7%	[123]
PVDF/PES	BSA (10 mg/L)	AOPs	100 kDa	Electrostatic repulsion	0.10	-	-	[124]
PVDFSMANa	BSA (500 mg/L)	Adsorption	100 kDa	Adsorption/Size retention/Electrostatic repulsion	0.10	1014	98.9%	[125]
PVDF	NOM	Coagulation	150 kDa	Adsorption/Size retention/Electrostatic repulsion	-	125	-	[126]

## Data Availability

Not applicable.

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
