# Peer review of "Reduction of Ultrafiltration Membrane Fouling by the Pretreatment Removal of Emerging Pollutants: A Review"

_membranes, 2023, doi:10.3390/membranes13010077_

Round 1

Reviewer 1 Report

 The topic is interesting and the whole manuscript is easy to read and understand. However, there are still some questions for the authors to clarify as follows:

1. Explain what you want to say ”Membrane pollution as a result of MPs mainly takes place...” Pg. 3, 

2. Subtitle 2.2, from pg. 5. What you what to say be Removal mechanism of pretreatment process? How to remove a process?

3. Coagulation/Flocculation, pg. 8. You can do a graphical illustration of different results obtained by other authors?

4. At pg. 11 Explain the environmental risk associated with pretreatment processes! How a pretreatment process affect the environment? You can explain how affect the treatment costs?

In the conclusions, please make more details of results!

  1. In the conclusions please make more details of results!

Reviewer 2 Report

This review presents the combination of pre-treatment operations of New Pollutants with the ultrafiltration process. 3 main pre-treatments have been identified: coagulation/flocculation, adsorption and advanced oxidative processes. A comparison of the effect of each process is made in relation to the filtration performance. This work is very relevant and interesting because these new pollutants have a great impact on humans in the field of water treatment. We could have expected a little more information on the effect of pre-treatments on the fouling of UF membranes and the different associated cleaning techniques. Moreover, there are very few comments related to the modification of the structure of the membranes, only a few works refer to this. Finally, I would recommend to make an introduction of each part between the paragraphs to help the reader, but also to create figures or a summary table summarizing the interactions between pre-treatment filtration performance and clogging proposed solutions. In my opinion, table 1 is not sufficient to describe all the paragraphs presented.

My comments :

Introduce each part if possible to help the reader understand the review. We often go from one part to another without having a link.

For each major part, one could have expected a figure that summarizes the main comments and conclusions. The review is often limited to rewriting short summaries of the publications without extracting the main information. The authors should try to highlight the key points of each work.

Generally, authors do not discuss the different operations to degrade these new pollutants afterwards. Once concentrated or trapped in the membrane, what is the fate of these PMs? What does the literature propose in this sense? This questioning would be a significant addition to the review.

Introduction

The introduction lacks size or molecular weight values for the various micropollutants presented in this study. The authors should add this to complete their comments.

L63: What about membrane nature, ceramic or polymer, on EPs retention?

L67 : membrane pollution would refer to degradation of the membrane or the internal external fouling of the membrane. The authors can also ask the question of the degradation of membranes in microplastic particles. Could the authors complete the comment?

L88: It is difficult to appreciate the relevance of figure 1. The authors should describe how it works and its usefulness for the review.

L125: Figure 2 is difficult to read. The text is too small which makes it impossible to understand. It needs to be redesigned.

L125 126 127: there seems to be a contradiction in the comments. Can the authors comment?

L134: for people who are not in the field, the unit kbp and its definition could be introduced

L139: refer to L67, authors should clearly say what they mean by pollution.

L142: which metal ion ?

Summarize the first paragraph and introduce the next one

A151: Chemical products are used. What is their environmental impact? Can the authors discuss?

Figure 3 is a bit unclear. Also, what does CUF mean? Could the authors make the figure a little more explicit and highlight key points?

L179 180: can authors give examples of which processes they figure out? Is it possible to classify those processes?

Presentation or introduction of the table 1 should be written

Table 1 : authors should introduce Transmembrane pressure.

L200 : Eps changed by EPs

L211 : what is eARGs?

L234 : Eps changed by EPs

L253 254 309 : EPS by EPs

L319: what authors mean by photolysis and photocatalysis? examples should be given

L414: in this paragraph the authors should comment on the effect of pre-treatment on membrane clogging and backwashing steps. There are repetitions with the previous paragraphs which seem to me to bring nothing new.

Round 2

Reviewer 1 Report

The paper can be accepted.